# Schedule-Dependent Treatment Increases Chemotherapy Efficacy in Malignant Pleural Mesothelioma

**DOI:** 10.3390/ijms231911949

**Published:** 2022-10-08

**Authors:** Darya Karatkevich, Haibin Deng, Yanyun Gao, Emilio Flint, Ren-Wang Peng, Ralph Alexander Schmid, Patrick Dorn, Thomas Michael Marti

**Affiliations:** 1Department of General Thoracic Surgery, Inselspital, Bern University Hospital, Murtenstrasse 28, 3008 Bern, Switzerland; 2Oncology-Thoracic Malignancies, Department of BioMedical Research, University of Bern, 3010 Bern, Switzerland; 3Graduate School of Cellular and Biomedical Sciences, University of Bern, 3010 Bern, Switzerland

**Keywords:** pemetrexed, cisplatin, chemotherapy, schedule-dependent treatment, malignant pleural mesothelioma, DNA damage, senescence

## Abstract

Malignant pleural mesothelioma (MPM) is a rare but aggressive thoracic malignancy with limited treatment options. One of the standard treatments for MPM is chemotherapy, which consists of concurrent treatment with pemetrexed and cisplatin. Pemetrexed limits tumor growth by inhibiting critical metabolic enzymes involved in nucleotide synthesis. Cisplatin causes direct DNA damage, such as intra-strand and inter-strand cross-links, which are repaired by the nucleotide excision repair pathway, which depends on relatively high nucleotide levels. We hypothesized that prolonged pretreatment with pemetrexed might deplete nucleotide pools, thereby sensitizing cancer cells to subsequent cisplatin treatment. The MPM cell lines ACC-MESO-1 and NCI-H28 were treated for 72 h with pemetrexed. Three treatment schedules were evaluated by initiating 24 h of cisplatin treatment at 0 h (concomitant), 24 h, and 48 h relative to pemetrexed treatment, resulting in either concomitant administration or pemetrexed pretreatment for 24 h or 48 h, respectively. Multicolor flow cytometry was performed to detect γH2AX (phosphorylation of histone H2AX), a surrogate marker for the activation of the DNA damage response pathway. DAPI staining of DNA was used to analyze cell cycle distribution. Forward and side scatter intensity was used to distinguish subpopulations based on cellular size and granularity, respectively. Our study revealed that prolonged pemetrexed pretreatment for 48 h prior to cisplatin significantly reduced long-term cell growth. Specifically, pretreatment for 48 h with pemetrexed induced a cell cycle arrest, mainly in the G2/M phase, accumulation of persistent DNA damage, and induction of a senescence phenotype. The present study demonstrates that optimizing the treatment schedule by pretreatment with pemetrexed increases the efficacy of the pemetrexed-cisplatin combination therapy in MPM. We show that the observed benefits are associated with the persistence of treatment-induced DNA damage. Our study suggests that an adjustment of the treatment schedule could improve the efficacy of the standard chemotherapy regimen for MPM and might improve patient outcomes.

## 1. Introduction

Malignant pleural mesothelioma (MPM) is a very rare and aggressive cancer, with etiology being heavily linked to previous asbestos exposure. The treatment of MPM is complex. In detail, in patients with early-stage MPM, the role of radical surgery is still debatable and should only be considered as part of a multimodality approach. For non-resectable MPM, there have been no significant changes after the approval of antifolate and platinum combination chemotherapy [1]. The FDA recently recommended immunotherapy as a first-line treatment for unresectable pleural mesothelioma. It combines two immune checkpoint inhibitors: CTLA-4 (ipilimumab) and PD-1 inhibitors (nivolumab and pembrolizumab). Clinical trials have shown high therapeutic effects. Due to contraindications or side effects, not all patients with mesothelioma are eligible for immunotherapy. In this case, chemotherapy will be the first option to choose.

The median overall survival with the standard first-line chemotherapy treatment is approximately 13 months, with the best results for the patients with the epithelial histological subtype of mesothelioma [2]. It has already been shown that the scheduled administration of different treatments can increase the efficacy of standard therapy. In a non-small cell lung cancer model, our group recently showed that pretreatment with pemetrexed (multitargeted antifolate, MTA; commercial name ‘Alimta’) followed by radiation therapy results in persistent DNA damage and induction of cellular senescence compared with standard treatment [3]. Similar molecular changes were observed after 48 h of pretreatment with MTA, followed by cisplatin [4].

Drugs targeting pyrimidine and purine biosynthesis, the basic building blocks of DNA, have been used in cancer therapy for several decades. MTA limits tumor growth by inhibiting three metabolic enzymes, e.g., the folate-dependent enzymes thymidylate synthase (TS) and dihydrofolate reductase (DHFR), and the purine biosynthetic enzyme, glycinamide ribonucleotide formyltransferase (GARFT) [5]. Thus, treatment with MTA leads to reduced biosynthesis of purines and pyrimidines, thereby inducing an imbalance in the nucleotide pool, and subsequently to the formation of single-stranded DNA at replication forks, which, if not repaired, can lead to the formation of double-stranded DNA breaks [6,7,8,9]. Cisplatin is an alkylating agent that causes direct DNA damage, such as intra-strand and inter-strand DNA cross-links, which are repaired by nucleotide excision repair, which is dependent on sufficiently high nucleotide levels [10].

The formation of DNA double-stranded breaks (DSBs) activates the DNA damage response signaling cascade leading to the phosphorylation of the histone γH2AX—a key mediator for the assembly of DNA repair proteins at the sites of DNA damage as well as for the activation of checkpoint proteins. Therefore, the analysis of γH2AX is often used as a marker for DNA damage response activation [11,12]. If not repaired, persistent DNA damage leads to the induction of senescence, e.g., an irreversible cell cycle exit associated with the conversion to an immunogenic phenotype that facilitates self-elimination by the immune system [13].

In this study, we aim to demonstrate the superiority of schedule-dependent versus concomitant cisplatin/MTA combination therapy in MPM. We have performed in-depth analyses at the molecular and cellular levels to investigate the underlying mechanisms of schedule-dependent drug administration. We were able to show that a 48 h pretreatment with MTA and subsequent cisplatin therapy results in increased cell cycle arrest, accumulation of persistent DNA damage, and induction of senescence compared to standard concomitant therapy.

## 2. Results

### 2.1. Schedule-Dependent Treatment Regimen Increases the Efficacy of MTA-Cisplatin Therapy

Three different treatment regimens were compared to determine whether the treatment schedule influenced the effectiveness of the MTA-cisplatin combination therapy. The three regimens consisted of continuous MTA (1 μM) treatment for 72 h (Figure 1A) in combination with cisplatin (5 μM) treatment for 24 h at different intervals. In detail, cisplatin was applied at 0 → 24 h (concomitant treatment), 24 → 48 h (24 h MTA pretreatment), or 48 → 72 h (48 h MTA pretreatment) relative to the 72 h of MTA treatment (Figure 1A). The doubling time (day 0 → 3) of untreated MESO-1 cells was approximately 22 h (Figure 1B), which is in agreement with the information provided by the American Type Culture Collection. In the short term, the two pretreatment strategies (24- and 48 h pretreatment) significantly reduced cell counts compared to the untreated control (Figure 1B). The highest efficacy in the initial phase of treatment was shown by concomitant cisplatin-MTA treatment (Figure 1B, left top panel), which was also observed after treating H28 cells with same treatment regimens (Figure 1B, left bottom panel). For all three regimens, cell proliferation was reduced in both cell lines already after the first day of treatment. After the treatment phase (day 4), MESO-1 cells featured an increased cell proliferation rate after both pretreatment regimens compared to concomitant therapy (Appendix A).

To estimate the long-term growth capacity of the remaining cells and the efficacy of the treatment regimen, MESO-1, and H28, cells were harvested on day 10 and reseeded at low density. Three days after reseeding, e.g., on day 13, cell numbers of both cell lines were significantly reduced by 48 h of MTA pretreatment compared to the alternative treatment regimens (Figure 1B, right panels). Between day 13 and day 20, the proliferation rate of MESO-1 cells after concomitant therapy was higher compared to 24 h and 48 h MTA pretreatment (Appendix A). From day 17, after 24 h of MTA pretreatment and concurrent therapy, the cultures of both cell lines started to reach a plateau phase due to high cell density, which persisted until day 24. The cultures of MESO-1 and H28 cells did not reach the plateau phase after 48 h of MTA pretreatment until the end of the experiment.

In conclusion, at the end of the short-term assay (day 10), the concurrent treatment showed the highest growth inhibitory effect compared with both pretreatment regimens in both MPM cell lines. However, compared with the other two treatment regimens, the 48 h MTA pretreatment significantly reduced the long-term growth capacity of MPM cells.

### 2.2. 48 h Pretreatment with MTA Induces Cellular Senescence

Visual examination revealed that a significant fraction of the cells had changed their morphology after treatment, namely, increased cell size and granularity, which is reminiscent of the induction of cellular senescence, as we have shown previously [3,4,9]. To confirm the induction of senescence, cells were reseeded at low density 6 days after therapy (day 10, see Figure 1B). Quantification of senescence-associated β-galactosidase (SA-β-Gal) activity five days later showed that about 30% of the cells featured a senescent phenotype after concomitant chemotherapy compared to 40% and 70% after 24 h pretreatment and 48 h pretreatment with MTA, respectively (Figure 2A,B).

The acquisition of a senescence phenotype is also associated with an increase in cell size and granularity (reviewed in [14]), which can be quantified by flow cytometry as an increase in the intensity of forward (FSC) and side scatter (SSC), respectively [9,15]. Indeed, the fraction of cells characterized by increased FSC and SSC intensity (F/S-high) was already maximally increased during the treatment phase of concomitant therapy (day 3), whereas the peak in the frequency of F/S-high cells was only reached on day 7 and day 10 after 24 h and 48 h of MTA treatment, respectively (Figure 2C,D). On day 17 of the recovery phase, the highest amount of F/S-high cells was observed after 48 h of pretreatment. Specifically, the percentage of cells with an F/S-high phenotype was 26.4% and 33.9% after concomitant therapy and 24 h pretreatment, respectively, compared with 45.6% after 48 h MTA pretreatment. In summary, compared with the other two treatment regimens, the 48 h MTA pretreatment significantly increased the fraction of senescent cells during the long-term recovery phase.

### 2.3. Prolonged MTA Pretreatment Enhances Cisplatin-Induced Cell Cycle Arrest

Senescence is associated with a permanent cell cycle arrest (reviewed in [13,14,16]). To understand the molecular mechanisms underlying the increased senescence induction by prolonged MTA pretreatment, we monitored cell cycle progression by flow cytometry. Previously, we observed that the morphological changes associated with senescence, e.g., increasing cell size and granularity, lead to a spillover signal into the DAPI channel during the analysis of the cell cycle by flow cytometry [9,15]. Indeed, others have also shown that the increase in the cell size of cancer cells after cisplatin treatment leads to a shift in DAPI signaling [17]. Therefore, in our FACS gating strategy, we first identified subpopulations by a high or low FSC/SSC signal intensity (F/S-low and F/S-high, respectively) and subsequently determined their cell cycle distribution individually based on DAPI staining (Appendix A), as we described previously [9,15].

Combined MTA-cisplatin treatment for 24 h (concomitant therapy, day 1–day 2) significantly reduced the fraction of F/S-low cells and resulted in this population in a dramatic S-phase arrest compared to the untreated MESO-1 cells (40% and 16%, Figure 3, top left panel). In contrast, cells from the F/S-high population were predominantly arrested in the G2/M phase after the same treatment (Figure 3, top right panel). The fraction of F/S-low cells remained unchanged 24 and 48 h after single MTA pretreatment compared to the untreated control (81%, 82%, and 85%, respectively). However, a single treatment with MTA for 24 h (day 2, both pretreatments) led to an S-phase arrest in F/S-high cells (around 25% of cells were arrested compared to 15% in untreated samples). Pretreatment with MTA for 48 h (day 3) did not significantly affect the cell cycle distribution of F/S-low cells. However, the S-phase arrest of F/S-high cells was further exacerbated and accompanied by an increase of cells arrested in the G2/M-phase, resulting in a G1-phase depletion (Figure 3, bottom right panel, day 3).

After both pretreatments, the addition of cisplatin resulted in F/S-low cells in a significant G1-phase depletion, consequently resulting in the accumulation in S/G2/M-phase, which was also observed in F/S-high cells after 24 h of MTA pretreatment. In contrast, after 48 h of MTA pretreatment, adding cisplatin for 24 h increased the fraction of F/S-high cells in the G1-phase.

During the recovery phase after concomitant treatment, the proportion of F/S-low cells increased continuously until day 20 (76%, see also Figure 2), and this subpopulation exhibited a near-normal cell cycle distribution (Figure 3, top left panel). The growth curve after concomitant therapy revealed that the cell culture reached high confluency on day 20 (Figure 1B, right panel). Indeed, on days 20 and 24 after concomitant therapy, F/S-low cells entered a G1-phase arrest, which was accompanied by an increase in the fraction of cells featuring an F/S-high phenotype, further corroborating that the culture reached its maximal confluency at these late time points.

Acquisition of a near-normal cell cycle distribution in F/S-low cells was delayed after 24 and 48 h of MTA pretreatment, as indicated by the presence of 25% and 39% of cells in the G2-phase at day 13 compared to 22% after concomitant therapy (Figure 3 left column).

On day 17, after concomitant treatment and 24 h of MTA pretreatment, 25% and 28% of the total cells retained an F/S-high phenotype, respectively (Figure 3 right side, upper and middle panels). In contrast, after 48 h of MTA pretreatment, 40% of the cells still featured an F/S-high phenotype (Figure 3, lower right panel). On the last day of the recovery phase, only 20% of F/S-high cells were in the G1-phase after 48 h of MTA pretreatment, whereas 40% and 65% of the cells were in the G1-phase after concomitant and 24 h MTA pretreatment, respectively (Figure 3, right panels).

Cell cycle analysis also highlighted the presence of a fraction of cells with sub-G1 DNA content, which is a hallmark of cells undergoing apoptosis [18]. In the F/S-low subpopulations (Figure 3, left panels), the fraction of cells featuring a sub-G1 DNA content was generally low. In the F/S-high subpopulations (Figure 3, right panels), few sub-G1 cells were detectable on days 10 and 13 after concomitant and 24 h of MTA pretreatment. The only treatment that resulted in 10% of sub-G1 cells (day 17) was 48 h of MTA pretreatment.

In summary, the different treatment regimens resulted in diverging cell cycle distributions. Interestingly, after 48 h of MTA pretreatment, the addition of cisplatin resulted in an increased fraction of F/S-high cells in the G1-phase. Recovery of the F/S-low fraction with a normal cell cycle distribution was maximally delayed after 48 h of MTA pretreatment, and the remaining F/S-high cells (day 24) were mainly arrested in the G2/M phase.

### 2.4. Prolonged MTA Pretreatment Results in the Accumulation of Persistent DNA Damage

It has already been shown that the accumulation of persistent DNA damage leads to cell cycle arrest and induction of senescence in A549 cells [19]. We compared the effect of the different treatment schedules using H2AX phosphorylation as a marker for DNA damage induction (Appendix A). During the treatment phase (day 1–4), H2AX phosphorylation was increased by all three different treatment regimens compared to the untreated control Figure 4). The highest level of H2AX phosphorylation was observed on day 3 after concomitant treatment. MTA treatment per se for 24 h did not significantly increase total H2AX phosphorylation during the initial 24 h (Figure 4, 24/48 h MTA pretreatments, day 1–2) but increased during the extended treatment period (Figure 4, 48 h MTA pretreatment, day 2–3). Interestingly, a drop in total H2AX phosphorylation was detectable for all treatment regimens on day 10 and day 20, e.g., once cultures reached relatively high confluency (Figure 4).

Upon DNA damage induction, H2AX phosphorylation is differentially increased depending on the cell cycle phase [12]. Thus, we combined the analysis of H2AX phosphorylation and cell cycle progression (Figure 5), as described by us before [3,4,9]. In detail, 48 h after single MTA treatment, 30% and 20% of total F/S-low cells in the S- and G2M-phase featured increased H2AX phosphorylation, which was only observed in 15% of the total population of the corresponding cells in the G1-phase. The same cell cycle-related pattern of H2AX phosphorylation was detectable in F/S-high cells (Figure 5, day3, left and right bottom panels, respectively).

H2AX was phosphorylated, 24 h after the start of the concomitant treatment, in the majority of F/S-low cells in the S- and G2/M-phase (Figure 5, top left panel, day 2), which was even more pronounced after 48 h. However, already 24 h later, i.e., still during the MTA treatment, H2AX phosphorylation started to decrease in F/S-low cells and continued to do so until almost basal levels were reached on day 10. In contrast, 48 h-pretreatment with MTA followed by 24 h of combined cisplatin and MTA treatment (day 4) did not immediately result in increased levels of H2AX phosphorylation in all phases of the cell cycle in F/S-low cells and F/S-high cells (Figure 5, bottom panels, day 4). However, on day 10, increased levels of H2AX phosphorylation in all cell cycle phases were observed in F/S-low and F/S-high cells after 48 h MTA pretreatment compared to the other treatment groups (Figure 5).

During the recovery phase (day 10–day 24), F/S-high cells featured high levels of H2AX phosphorylation after all treatment regimens. Remarkably, compared to the two alternative treatment regimens, H2AX phosphorylation levels were still increased in both F/S-low and F/S-high cells on days 20 and 24 after 48 h-pretreatment with MTA (Figure 5). In conclusion, 48 h of MTA pretreatment followed by 24 h of combined cisplatin and MTA treatment is the most effective modality to induce DNA damage that persists during a prolonged recovery phase in MPM cells.

## 3. Discussion

The current study provides for the first time a comparison and detailed analysis of schedule-dependent treatment variations with MTA and cisplatin in MPM cells over an extended recovery period. We found that prolonged MTA pretreatment combined with subsequent cisplatin therapy increases long-term DNA damage accumulation, which is associated with reduced survival, consistent with our previous findings in NSCLC [4]. A phase III clinical trial in MPM revealed that treatment with cisplatin alone was less effective than combination with MTA. To reduce the overall toxicity of MTA, supplementation with vitamin B12 and folic acid was undertaken. Importantly, no adverse effects of supplementation on treatment efficacy were observed [2]. Although an antagonistic effect was initially suspected, supplementation with vitamin B12 and folic acid was shown to actually increase MTA efficiency in several NSCLC cell lines in vitro [20]. Nevertheless, further experiments are needed to investigate how folate supplementation might affect a nucleotide depletion strategy in an immunocompetent tumor model system before applying a schedule-dependent treatment strategy in the clinical setting.

To our knowledge, a schedule-dependent effect of MTA-cisplatin combination therapy has never been demonstrated in a pleural mesothelioma model. In the context of MTA-based combination therapies, Nagai et al. demonstrated a schedule-dependent effect of the combination of MTA and gemcitabine for treating MPM and NSCLC. They found a decrease in cell viability for the MPM cell line MSTO-211H when MTA was administered for 24 h, followed by 4 h of Gemcitabine treatment [21]. Our previous work revealed that MTA pretreatment of NSCLC cells also increased the anti-cancer effect of subsequent radiotherapy [3]. Thus, our current study adds to the growing body of evidence that MTA pretreatment augments the anti-cancer effect of DNA damaging agents.

Our experiments revealed that prolonged MTA pretreatment leads to enhanced senescence induction in MPM cells (Figure 2), which is consistent with our previous results in NSCLC cells after subsequent treatment with cisplatin and radiotherapy [3,4]. Interestingly, MTA/cisplatin combination therapy significantly increased beta-gal activity in our study on the ACC-MESO-1 cell line and in a study by another group on an MPM primary culture [22], both containing a wild-type BAP1 gene. In the same study, a second primary culture with a BAP1 mutation featured elevated baseline levels of beta-gal activity, which was not further increased by MTA/cisplatin combination treatment. Thus, the induction of senescence by chemotherapy might be dependent on the genetic background of the MPM cells as suggested previously [22]. The exact role of senescence in therapy response is still under debate [23]. On one side, senescent cells feature a senescent associated secretory phenotype, which allows damaged cells to communicate their compromised state to the surrounding tissue thereby triggering a robust immune response [24]. On the other side, higher expression levels of senescence markers after treatment with neoadjuvant chemotherapy is associated with increased resistance and poor survival in a proportion of patients with MPM [25]. Currently, agents targeting senescent cells, e.g., senolytic drugs, are currently being evaluated with and without DNA-damaging agents in clinical studies for cancer therapy. Determining the optimal treatment schedule for therapies consisting of multiple DNA-damaging agents in combination with senolytic drugs will be of paramount importance.

Induction of senescence is associated with an increase in cell size and granularity, i.e., an F/S-high phenotype. Our previous results with A549 NSCLC cells showed that 48 h of MTA pretreatment followed by cisplatin therapy resulted in a steady increase in cells with an F/S-high phenotype, up to 96% of the total population on day 14 of the recovery phase. In contrast, in this study with MESO-1 MPM cells, the maximum increase (53%) in the proportion of F/S-high cells was reached as early as day 3 of the recovery phase and steadily decreased thereafter (increasing again on day 24 as the culture became fully confluent, see Figure 3). In our previous study testing different schedules of MTA and cisplatin combination therapies, we treated NSCLC cells with 1 μM MTA and 10 μM cisplatin [4]. In the current study focusing on MPM, we combined 1 μM MTA with 5 μM cisplatin as we used before to treat the MESO-1 cell line [26]. Although differences in tumor entity and genetic background could also influence the outcome, we speculate that the higher cisplatin dose resulted in a higher degree of persistent DNA damage and, thus, a higher proportion of senescent cells (F/S-high cells) in our NSCLC study. Concluding from the results of our two studies, we speculate that treating patients in the clinical setting with maximally tolerated doses of DNA-damaging agents may be beneficial for combination therapies with senolytic drugs in the adjuvant setting. However, in terms of further clinical translation, additional studies are needed that focus on how the tumor stage and changes in DNA repair mechanisms based on tumor-specific mutations affect the efficacy and safety of schedule-dependent combination strategies with DNA-damaging agents.

Finally, the overall effect of chemotherapy is limited not only by the induction of DNA damage response and cellular senescence but also by the activation of the immune system to eliminate tumor cells [27]. Indeed, it was shown that treatment with chemotherapy, particularly MTA, induces PD-L1 expression in various NSCLC cell lines by activating both mTOR/P70S6K and STAT3 pathways. The induction of PD-L1 expression by MTA might explain the observed activity of the MTA-based chemotherapy combined with pembrolizumab in PD-L1-negative NSCLC [28]. Moreover, for combining DNA-damaging therapy with immunotherapy, it will be pivotal to determine an optimal treatment schedule to maximize therapy efficiency and elicit durable immune responses to provide the most significant benefit to cancer patients.

The present study is consistent with our previous study in NSCLC, thereby corroborating our finding that the efficacy of MTA-cisplatin combination therapy can be increased in vitro by adjusting the treatment schedule. The observed higher efficacy of prolonged MTA pretreatment is due to persistent DNA damage accumulation leading to senescence induction. This study warrants further investigation on whether schedule-dependent treatment could increase the efficacy of combination chemotherapy in vivo. Finally, the introduction of schedule-dependent treatment strategies may have the potential to improve survival not only in mesothelioma but also in other tumor entities.

## 4. Materials and Methods

### 4.1. Cell Culture and Reagents

The MPM cell lines ACC-MESO-1 and NCI-H28 (hereafter MESO-1 and H28, respectively) were purchased from American Type Culture Collection (ATCC, Manassas, VA, USA). Cell lines were fingerprinted (Microsynth, Bern, Switzerland) and routinely tested for mycoplasma contamination using Mycoplasma Detection Kit (Cat. #rep-mys-10, InvivoGen, San Diego, CA, USA). Cell lines were cultured in Dulbecco’s modified Eagle’s medium nutrient mixture F12 Ham (Cat. #D6421, Sigma-Aldrich, St. Louis, MO, USA), supplemented with 10% fetal bovine serum (Cat. #10270-106; Life Technologies, Grand Island, NY, USA), 1% Penicillin/Streptomycin solution (Cat. #P0781, Sigma-Aldrich) and 1% L-Glutamine (Cat. #25030-024, Sigma-Aldrich) at 37 °C in a humidified 5% CO_2_ incubator. Medium was exchanged every 3 days. Cells were passaged using TrypLE as indicated by the manufacturer (Cat. #A1217702, Thermo Fisher Scientific, Waltham, MA, USA).

Pemetrexed/MTA (commercial name ‘ALIMTA’; Cat #VL7640) was purchased from Eli Lilly (Suisse) S.A. (Vernier/Geneva, Switzerland). Cisplatin (name ‘Cisplatin Ebewe’) was purchased from Sandoz Pharmaceuticals AG (Steinhausen/Cham, Switzerland).

### 4.2. Drug Response and Senescence-Associated β-Galactosidase Assay

On day 0, either 0.5 × 10^6^ MESO-1 cells or H28 cells were seeded into 150 mm × 20 mm plates with 25 mL of supplemented media. Then, 1 μM MTA was added on day 1 to all three regimens for a duration of 72 h. In the concomitant regimen, 5 μM cisplatin was added during the first 24 h of MTA treatment (0 h → 24 h). In the 24 h pretreatment regimen, 5 μM cisplatin was added during 24 and 48 h of MTA treatment (24 h → 48 h). In the 48 h pretreatment regimen, 5 μM cisplatin was added during 48 and 72 h of MTA treatment (48 h → 72 h). At the end of each 24 h-period of cisplatin treatment, media containing the drug was washed out once with PBS. Cell titers and cell viability were determined on days 1, 2, 3, 4, 7, and 10, respectively. Cells were harvested using TrypLE (Cat. #A1217702, Thermo Fisher Scientific, Waltham, MA, USA). Cell numbers were determined using a hemocytometer and 0.4% Trypan blue (Cat. #15250061, Thermo Fisher Scientific) (final concentration 0.1%) for dead cell exclusion. To prevent plates from reaching confluence, 0.5 × 10^6^ cells from each treatment regimen were reseeded on day 10 into novel 150 mm × 20 mm plates. Subsequently, long-term cell titers and viability were determined on days 13, 17, 20, and 24. MESO-1 cells from each treatment regimen were washed in phosphate-buffered saline, fixed at the indicated time points (day1 to day24, d1 → d24), and processed for analysis by flow cytometry as described below. All experiments were performed at least three times.

Senescent cells were visualized using the senescence β-galactosidase staining kit (Cat. #9860, Cell Signaling Technology, Danvers, MA, USA). In detail, 0.5 × 10^5^ MESO-1 cells on day 10 were reseeded in tissue culture treated 6-well plates (#Cat.343046, Corning, Inc., Corning, NY, USA). Cells were fixed after 5 days and stained overnight for β-galactosidase activity using the reagents provided by the kit and according to the manufacturer’s protocol. Visual quantification of senescent cells was performed by acquiring images of five random fields per treatment group using an inverted light microscope (EclipseTS100, Nikon Instruments Inc, Melville, NY, USA) equipped with a 20× objective. The number of senescent cells and total cells per field were counted using the ‘Cell Counter’ plugin of the ImageJ software. Experiments were repeated independently three times.

### 4.3. Flow Cytometry

Cell cycle distribution and DNA damage persistence were determined by flow cytometry based on our previously published protocols [3,4,7,9,12,29]. In detail, samples from all treatment regimens were collected as described above. Subsequently, 500,000 cells in each sample were washed with phosphate-buffered saline (PBS), pH 7.4, fixed with 0.5 mL IC Fixation Buffer (Cat. #00-8222-49, Thermo Fisher Scientific). Samples were stored at 4 °C for future staining. An untreated control was included for each time point. Cells were permeabilized with 0.1% Triton X-100 (Cat. #9036-19-5, Sigma-Aldrich, St. Louis, MI, USA). Staining with Alexa Fluor 488 anti-γH2AX (Ser139) (Cat. #613406, BioLegend, San Diego, CA, USA) antibody at a concentration of 250 ng/mL was performed in phosphate-buffered saline (Pharmacy, University Hospital Bern, BE, Switzerland) supplemented with 1% fetal bovine serum (Cat. #10270-106, Life Technologies, Grand Island, NY, USA) on a rotating wheel (3 rpm) overnight at 4 °C. The next day, cells were stained with 0.5 μg/mL 4′,6-diamidino-2-phenylindole (DAPI) (Sigma-Aldrich, Cat #D9542, St. Louis, MI, USA). Samples were pipetted through 35 μm nylon mesh filters into round-bottomed test tubes (Cat. 352235, Corning). Cell fluorescence was measured on an LSR2 flow cytometer (BD Biosciences), and data were analyzed using FlowJo V10 (Tree Star, Inc. Ashland, OR, USA). Samples from the different time points were stored at 4 °C, and flow cytometric analysis of all samples from one experiment was performed in parallel. An untreated control accompanied each analysis. Control samples were used to set the gating threshold for γH2AX positivity to ~10% of the total cell population as described by us before [3,14].

### 4.4. Statistical Analysis

Data were analyzed using Prism8 (GraphPad). In all studies, data represent biological replicates (n) and are depicted as mean values ± standard deviation (SD) or mean values ± standard error of the mean (SEM), as indicated in the figure legends. Comparison of mean values was conducted with two-tailed Student’s *t*-test, one-way and two-way ANOVA with Tukey’s multiple comparisons tests as indicated in the figure legends. In all analyses, *p* values less than 0.05 were considered statistically significant.

## Figures and Tables

**Figure 1 ijms-23-11949-f001:**
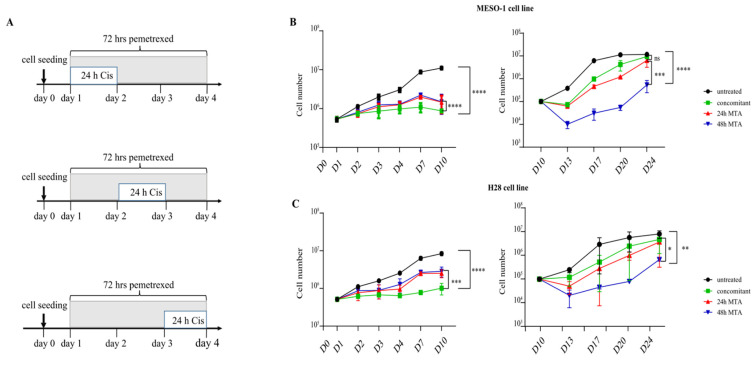
A schedule-dependent treatment plan improves the therapeutic efficacy of MTA-cisplatin combination therapy. (**A**) Schematic representation of the three treatment regimens, which differ in the duration of MTA pretreatment before the addition of cisplatin (see text for details). (**B**) Growth curves of MESO-1 (*n* = 3) cells during the treatment (d1–d4) and the recovery phase (d10–d24). Cells exposed to the indicated treatment regimen were harvested at day 10, reseeded, and cell numbers were determined at the indicated time points. Data represent means of three independent experiments, and bars indicate standard deviations. Two-way ANOVA was used to compare different treatment groups (ns *p* > 0.05 (not significant), and **** *p* < 0.0001). (**C**) Growth curves of H28 (*n* = 2) cells during treatment phase (d1–d4) and recovery phase (d10–d24). Two-sided student’s t test was used to analyze growth capacity at day 24 (* *p* < 0.05, ** *p* < 0.01, *** *p* < 0.001, **** *p* < 0.0001).

**Figure 2 ijms-23-11949-f002:**
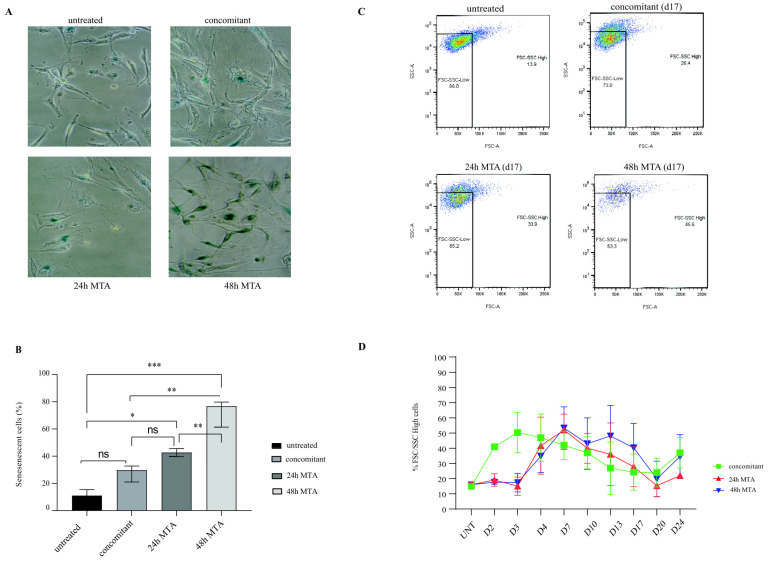
Prolonged MTA pretreatment exacerbates cisplatin-induced senescence induction. (**A**) Representative images of cell cultures taken by phase contrast-based microscopy after they were harvested on day 10 and allowed to regrow for 5 days. Blue staining indicates cells, which stain positive for senescence-associated β-galactosidase activity. (**B**) Quantification of senescent cells based on increased β-galactosidase activity. Data represent the means of three independent experiments, and bars indicate means and standard deviations. Two-way ANOVA was used to compare different treatment groups (ns p > 0.05 (not significant), * *p* < 0.05, ** *p* < 0.01, and *** *p* < 0.001). (**C**) Forward and side scatter analysis by flow cytometry at day 17 of the recovery phase. (**D**) Quantification of flow cytometry data over time. Data represent the means of three independent experiments, and bars indicate standard deviations.

**Figure 3 ijms-23-11949-f003:**
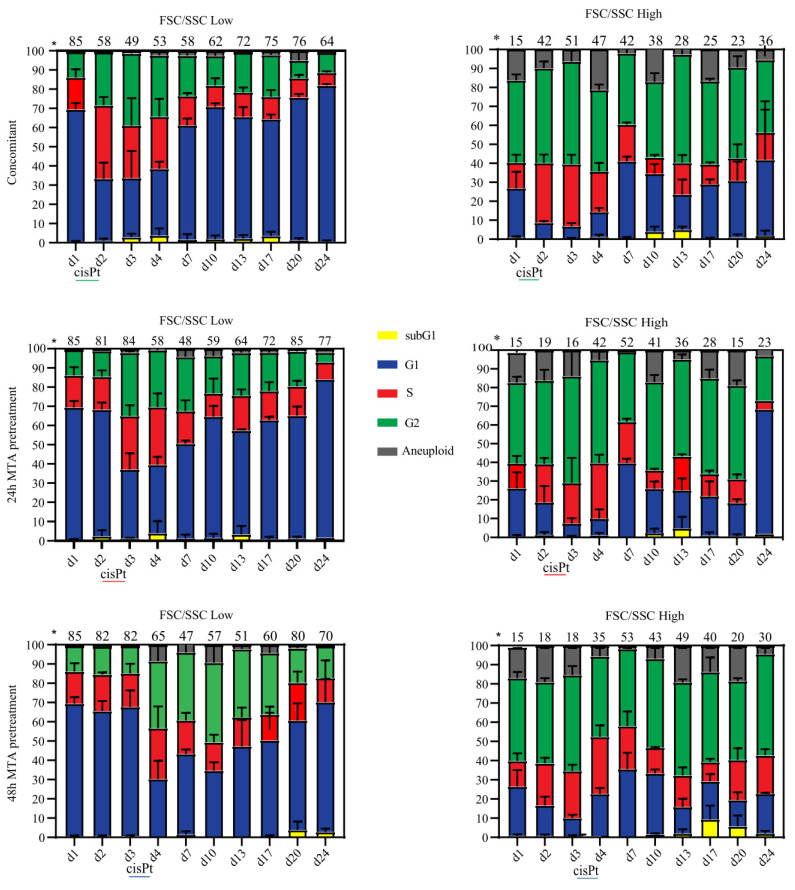
Prolonged MTA pretreatment enhances cisplatin-induced cell cycle arrest. Graphical representation of the cell cycle distribution from cell cultures after different treatment regimens, analyzed by flow cytometry at the indicated time points. Subpopulations featuring either increased forward and side scatter intensity (F/S-high) or low forward and side scatter intensity (F/S-low) were identified as indicated in Appendix A. Cell cycle analysis was performed as indicated in Appendix A. The data shown are the mean values and standard deviations of three experiments. * Percentage of the total cell population (mean from three experiments).

**Figure 4 ijms-23-11949-f004:**
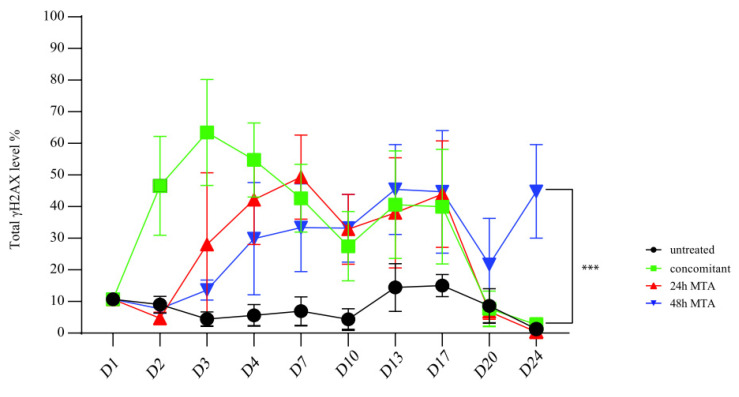
Prolonged MTA pretreatment enhances the cisplatin-induced accumulation of persistent DNA damage. Basal H2AX phosphorylation was set at ~10% in untreated control and used for normalization among experiments as described in the “Materials and methods” section. H2AX phosphorylation levels were determined in the whole population as described in Appendix A. One-way ANOVA test was performed to compare level of γH2AX between treatment groups at day 24 (***, *p* < 0.001).

**Figure 5 ijms-23-11949-f005:**
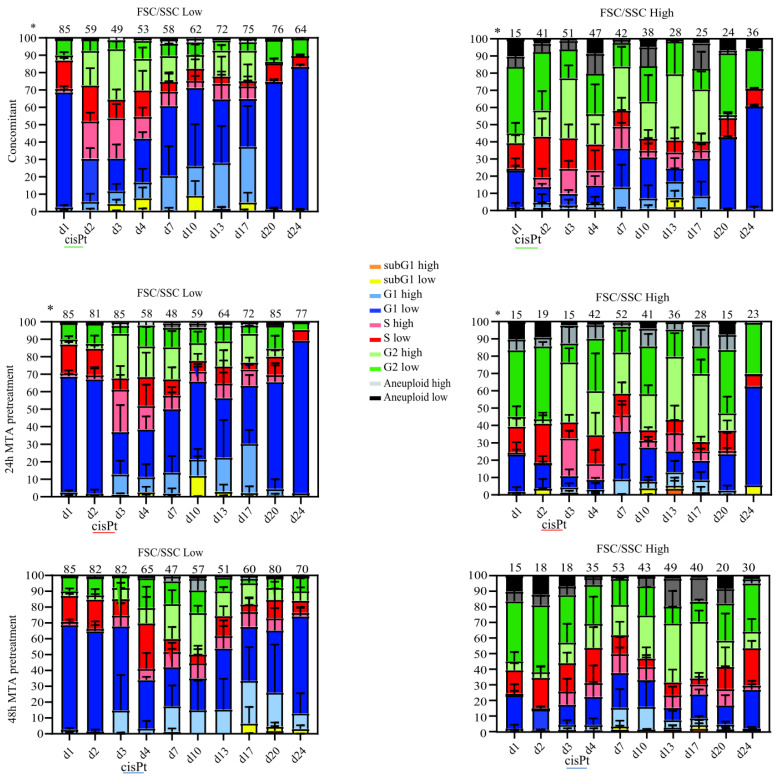
The cell cycle arrest induced by prolonged MTA pretreatment is associated with the accumulation of persistent DNA damage. Basal H2AX phosphorylation level was set at 10% in untreated control and further used for normalization during the experiment as described in the material and methods section. Cell cycle phase-specific H2AX phosphorylation levels were determined by combining the subpopulation–specific γH2AX and cell cycle gates, respectively (Appendix A). Data shown are the mean values and standard deviations of three independent experiments. * Percentage of the total cell population (mean from three experiments). Low = γH2AX negative, high = γH2AX positive.

## Data Availability

All data generated or analyzed during this study are included in this published article and its Appendix A.

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
