# Peer review of "Schedule-Dependent Treatment Increases Chemotherapy Efficacy in Malignant Pleural Mesothelioma"

_ijms, 2022, doi:10.3390/ijms231911949_

Round 1
Reviewer 1 Report
I really appreciate this manuscript and I think that it could deserve pubblication.
I only suggest to Authors to add some considerations about the potential translational value of their study, trying to highlight which could be future applications to clinics of these observations.
I think that this additional considerations would add value and interest for a broader set of readers.
Reviewer 2 Report
The authors speculated that prolonged pretreatment with pemetrexed might deplete nucleotide pools, thereby sensitizing cancer cells to subsequent cisplatin treatment in pleural mesothelioma. Experimental work, well conducted, clear in the discussion of the results.
Reviewer 3 Report
General:
Karatkevich et al. present a well written and excellent manuscript on the possible improvement of the combination chemotherapy pemetrexed/cisplatin for patients with malignant pleural mesothelioma (MPM). This is an important topic because currently the only alternative for first-line treatment of unresectable MPM, besides standard chemotherapy, is immunotherapy – which is quite expensive and not suitable for all patients.
There are only minor, mostly formal, issues that should be addressed.
Minor:
1) Patients who are treated with pemetrexed are given vitamin B supplements (B12 and folic acid) to reduce side effects. Please add one or two sentences to the discussion whether or how this might affect your nucleotide depletion strategy when applying the schedule-dependent treatment to patients in the future.
2) If early detection of MPM would be possible, do you see any synergistic effects when applying your schedule-dependent treatment to early-stage tumors?
line 73: “activates” instead of “activated”
line 94: “CA” instead of “California”.
line 115: Why is “Waltham, MA” underlined (linked to a google search)?
line 150f: “(Tree Star, Inc., Ashland, OR, USA)” instead of “(Tree Star, Inc. (Ashland, OR, USA))”
Fig. 1 (& Fig. 2) and relating text: In the figures you use upper case letters (A, B, C) but in the main text and legends lower case letters (a, b, c). Please make it consistent.
line 177: “(Fig. 1c, left bottom panel)” instead of “((Fig. 1b, left bottom panel)”
line 196: “(Fig. 1b and 1c, right panels)” instead of “(Fig. 1b, right panels)”
line 249: “them” can be deleted
line 422: The sentence is incomplete.
line 460ff: Please check the format of ref. 1.
line 539: “Supplementary Fig. S3” instead of “Supplementary 3”
line 543: “Supplementary Fig. S4” instead of “Supplementary 4”
